# Mixed-Precision Computing in the GRIST Dynamical Core for Weather and Climate Modelling

Siyuan Chen[1,5], Yi Zhang[1,3,5], Yiming Wang[1,5], Zhuang Liu[2], Xiaohan Li[2], Wei Xue[4]

1 2035 Future Laboratory, PIESAT Information Technology Co., Ltd., Beijing, China

5  2 Ministry of Education Key Laboratory for Earth System Modelling, Department of Earth System Science, Tsinghua University, Beijing, China

3 Chinese Academy of Meteorological Sciences, Beijing, China

Department of Computer Science and Technology, Tsinghua University, Beijing, China

Beijing Research Institute, Nanjing University of Information Science and Technology, Beijing, China

*Correspondence to*: Yi Zhang (zhangyi_fz@piesat.cn)

**Abstract.** Atmosphere modelling applications become increasingly memory-bound due to the inconsistent development rates between processor speeds and memory bandwidth. In this study, we mitigate memory bottlenecks and reduce the computational load of the GRIST dynamical core by adopting the mixed-precision computing strategy. Guided by a limited-degree of iterative development principle, we identify the coded equation terms that are precision insensitive and modify them from double- to single-precision. The results show that most precision-sensitive terms are predominantly linked to

pressure-gradient and gravity terms, while most precision-insensitive terms are advective terms. Without using more computing resources, computational time can be saved, and the physical performance of the model is largely kept. In the standard computational test, the reference runtime of the model's dry hydrostatic core, dry non-hydrostatic core, and the tracer transport module is reduced by 24%, 27%, and 44%, respectively. A series of idealized tests, real-world weather and climate modelling tests, were then performed to assess the optimized model performance qualitatively and quantitatively. In

particular, in the long-term coarse-resolution climate simulation, the precision-induced sensitivity can form at the large scale. While in the kilometre-scale weather forecast simulation, the model sensitivity to the precision level is mainly limited to the small-scale features, and the wall-clock time reduces 25.5% from the double- to mix-precision full-model simulations.

## 1 Introduction

Increasing model resolution is an effective approach of enhancing the atmosphere model forecast accuracy (Bauer et al.

2021; Benjamin et al.2019; Yu et al. 2019). Highly accurate, efficient, stable and scalable global dynamical cores have been widely pursued over the past two decades (e.g., Tomita and Satoh 2004; Harris and Lin 2012; Skamarock et al. 2012; Zängl et al. 2015; Wedi et al. 2020; Sergeev et al. 2023; Zhang et al. 2023). Doubling the horizontal resolution with a fixed vertical resolution leads to an increase of computational amount by a factor of $\sim 2^3$, a significant challenge in terms of computational cost and energy consumption.

Operational weather and climate forecasting is a field where the dual demands of accuracy and computational efficiency converge, necessitating both quality and speed. In the context of high-resolution meso-scale forecasting, which operates on scales of a few kilometres, computational efficiency itself implies forecast accuracy. Faster models enable more frequent forecast-assimilation cycles and the use of larger ensemble sizes within the constraints of finite computational resources. To tackle these computational hurdles, efforts have concentrated on enhancing the efficiency of numerical models. Progress such as Field Programmable Gate Arrays (FPGAs) and heterogeneous computing (e.g., Gan et al. 2013; Yang et al. 2016; Fu et al. 2017; Gu et al. 2022; Taylor et al. 2023), alongside compiler optimizations (e.g., Santos et al. 2024), have demonstrated significant potential in accelerating earth system models.

Conventional weather/climate model development has typically relied on double-precision (64-bit) floating-point. The transition from double- to single-precision (32-bit) or even half-precision floating-point arithmetic presents an intriguing avenue for enhancing the computational efficiency (Düben et al. 2014). Single-precision computation unveils several compelling advantages, especially when confronted with the memory wall (Abdelfattah et al. 2021; Fornaciari et al. 2023; Brogi et al. 2024). Beyond the alleviation of memory constraints, single-precision arithmetic promises three distinct benefits: accelerated arithmetic operations, improved cache hit rates, and reduced inter-node data communication (Baboulin et al. 2009; Düben and Palmer 2014; Düben et al. 2015; Váňa et al. 2016; Nakano et al. 2018). The benefits highlighted illustrate the capability of single-precision computation to boost computational efficiency in high-performance computing tasks, especially within the realm of large-scale weather and climate simulations where computational expenses are significant.

However, a wholesale migration from double- to single-precision computing may not always yield beneficial outcomes. This has led to the exploration of precision sensitive model components and/or physical scales in earth system modeling (e.g., Thornes et al. 2017; Nakano et al. 2018; Chantry et al. 2019; Maynard and Walters 2019; Cotronei and Slawig 2020). Single-precision algorithms may struggle to converge or achieve the required precision when tackling intricate fluid dynamics simulations. In certain scenarios, single-precision computations can also result in floating-point under/overflow (Váňa et al. 2016; Cotronei et al. 2020). Additionally, physical parameterization schemes in the atmospheric models may amplify the grid-scale oscillations when executed in a pure single-precision mode (Váňa et al. 2016). Therefore, it becomes imperative to identify the specific algorithms within the modeling framework that are sensitive to the precision level.

Previous studies have made notable progresses. A pivotal study (Váňa et al. 2016) explored the reduction of almost all real-number variables in the Integrated Forecast System (IFS) of the European Center for Medium-Range Weather Forecasting (ECMWF) from 64 bits to 32 bits. Results revealed that reducing precision did not significantly compromise the model accuracy, while it considerably reduced the computational burden by a factor of ~40%. Based on the dynamical core of the nonhydrostatic icosahedral model (NICAM), Nakano et al. (2018) witnessed an undesirable wavenumber-5 structure when completely using single-precision computing. This abnormal wave growth was traced back to the errors in the grid cell geometry calculations. By using double precision for only necessary parts in the dynamical core and single precision for all other parts, the model successfully simulated the baroclinic wave growth, and achieved a ~46% reduction of runtime. Based

on the Yin-He global spectral model, Yin et al. (2021) used a single-precision fast spherical harmonic transform to conduct a 10-day global simulation and a 30-day retrospective forecasting experiment. Their simulations reproduced the major precipitation events over southeastern China. The single-precision fast spherical harmonic transform may lead to a reduction of runtime by ~25.28% without significantly affecting the forecasting skill. Cotronei et al. (2020) converted the majority of the computations within the radiation component of European Centre Hamburg Model (ECHAM) to single precision, resulting in a 40% reduction in the runtime of the individual component. The obtained results were comparable to those achieved with double precision. Banderier et al. (2023) indicated that employing single precision for regional climate simulations can significantly reduce computational costs (~30%) without significantly compromising the quality of model results.

While these studies have demonstrated various ways for precision optimization, certain limitations remain. First, some studies focused on a complete transition to single-precision, potentially overlooking the precision-sensitive components and lacked a discussion of optimization strategies. Moreover, the applicability of mixed precision in global climate simulations remains to be validated. Furthermore, because of the diversity of numerical models and algorithms, encompassing grid systems and solver techniques, these differences may lead to the model-specific precision sensitivity. Certain algorithms may remain amenable to single-precision computations, while others necessitate the use of double precision for stability and accuracy. These gaps in the literature underscore the need for the present research to explore precision sensitivity, and to test the reduced-precision computing for both weather and climate simulations.

In this study, we explored the strategies of mixed-precision computing in the dynamical core of the Global-Regional Integrated Forecast System (GRIST; Zhang et al. 2019; Zhang et al. 2020). GRIST is a unified weather-climate model system designed for both research and operational modeling applications. Through a detailed implementation by modifying certain parts of the original (double-precision) dynamical core to support single-precision, a significant reduction of the computational burden has been achieved without sacrificing the solution accuracy, stability, and physical performance. This has been validated based on a series of numerical tests ranging from idealized to real-world flow.

The remainder of this paper is organized as follows. Section 2 introduces the GRIST model, presents the mixed-precision optimization strategies, code modifications, and highlights the key equation terms sensitive to precision. Section 3 examines the computational performance of mixed-precision computing. Section 4 evaluates the physical performance of mixed-precision computing in a series of test cases. Discussion and conclusion are given in Sections 5 and 6, respectively.

**2.1. GRIST**

The GRIST dynamical core employs layer-averaged governing equations based on the generalized hybrid sigma-mass vertical coordinate and a horizontal unstructured grid, allowing a switch between the hydrostatic and non-hydrostatic solvers (Zhang 2018; Zhang et al. 2019; Zhang et al. 2020). Prognostic variables are arranged in a hexagonal Arakawa-C grid approach. The hydrostatic solver is fully explicit, based on the Runge-Kutta integrator and the Mesinger forward-backward scheme. The nonhydrostatic solver employs a horizontally explicit vertically implicit approach. There is no time splitting in

the integration of the dry dynamical core (dycore hereafter), while the tracer transport module is time-splitted from the dycore, and supports several transport schemes for various applications (Zhang et al. 2020). In this study, a third-order upwind flux operator combined with the Flux-Corrected Transport limiter is used in the horizontal, and an adaptively implicit method is used in the vertical (cf., Li and Zhang 2022).

### 2.2 Mixed-precision optimization strategy

The purpose is to decrease the precision level (and thus computational cost) and maintain the accuracy and stability. Before implementing mixed-precision computing, we have checked that completely using single precision for the entire dynamical core leads to an unacceptable accuracy loss (see Section 4.1). However, considering the extensive codebase and its degree of complexity, comprehensively and randomly testing every component and variable is impractical. An iterative development approach with a minimum degree of trial and error is used to identify the model components that are sensitive to the precision level. The dry baroclinic wave of Jablonowski and Williamson (2006) is used as a benchmark test during the iterative development cycle because this case has complex fluid dynamics characteristics and is very sensitive to numerical precision.

We established an acceptable error threshold, $\alpha$, to assess whether the difference between outcomes from double-precision and mixed-precision simulations falls within a tolerable limit. Results from original double-precision computing serve as the true values. The iteration is: initially, a 10-day simulation was executed. We then embarked on a series of precision reduction tests for selected model variables, and we computed the error norm of selected diagnostic variables for each test. $E$ is defined to represent the overall error level (calculated relative to the double-precision results):

$$E = \max\big(L(\mathcal{H})\big) \tag{1}$$

$$L(x) = \max\big(L_1(\mathcal{H}), L_2(\mathcal{H}), L_\infty(\mathcal{H})\big) \tag{2}$$

where $L_1$, $L_2$, and $L_\infty$ represent the first, second and infinite norm of variable $\mathcal{H}$. The definitions of $L_1$, $L_2$, and $L_\infty$ can be found in the appendix. Should "error" $E$ exceed $\alpha$ (0.05 for this study), the modification is deemed unacceptable and consequently abandoned; otherwise, the modification is accepted, allowing a further reduction in variable precision based on this new configuration. This is an optimization approach similar to the greedy algorithm. Initially, in selecting single-precision variables, we systematically attempted to reduce the precision of variables encountered sequentially in the code, starting with the first variable, followed by the second, third, and so forth. The precision optimization tests were conducted using the G8 grid. The grid names and their corresponding resolutions are in Table 1. Initially, selected diagnostic variables ($\mathcal{H}$) are $ps$ (surface pressure) and $vor$ (relative vorticity) because they can effectively quantify deviations in the mass field and velocity field. This criterion was set beforehand, while it has turned out that $L(vor)$ overall has much larger error magnitude than $L(ps)$. Thus, it is $L(vor)$ determines our optimization outcome.

Technically, the switch between double-precision and single-precision code is defined through the FORTRAN KIND parameter, specified in a constant module. As single-precision results may not always replicate the double-precision results and can occasionally generate unacceptable errors (e.g., see Section 3.2), it is crucial to identify precision-sensitive variables

and solver components. An additional parameter 'ns' has been introduced in this constant module for the precision-insensitive variables. This modification facilitates the transition between double-precision, single-precision, and mixed-precision computations. Please note that only the subroutine of the solver is modified, indicating that the model initialization section remains in double-precision operations. If the solver requires single-precision operands, double-precision variables need to be converted to single-precision after initialization. This method ensures a streamlined transition to mixed precision with minimal changes to the code structure.

Some important aspects are summarized as follows:

(i)  Model variables insensitive to the precision level are set to the type parameter "ns". When "ns" is defined as single precision, the code executes mixed-precision computations; when defined as double precision, the code regresses double-precision computing and produces identical solutions as the original unmodified code;

(ii)  Appropriately decompose computations involving implicit type conversions to reduce performance degradation due to precision conversion. For instance, 'a = b * c'. Here, 'a' is a single-precision floating point, 'b' a larger double-precision float, and 'c' a single-precision float. The conversion of 'c' to double-precision can introduce extra rounding errors. These errors, amplified by 'b', may accumulate over time, adversely affecting model outcomes. Single-precision calculations provide a consistent error boundary, unlike mixed-precision which introduces uncertainty. In some cases, results might even be better if the computation of a function were entirely in single precision. Hence, optimization should proceed with caution, considering these error dynamics.

(iii)  The Message Passing Interface (MPI) communication was modified for single-precision variables; The built-in functions such as 'HUGE' or 'TINY' are used to obtain very large or very small values respectively, to ensure the values fall within the precision range of the variables.

**2.3 Mixed-precision optimization results**

Following the strategy outlined in Section 2.2, the mixed-precision GRIST dynamical core is established. The optimization results, as depicted on the left side of Fig. 1, are summarized based on the continuous-form governing equations. The meaning of each variable in the equations *exactly* follows Zhang et al. (2020) so that we avoid repeating explanation. Model variables with underlined text denote single-precision operands, variables in black represent double-precision operands. Black dashed boxes indicate that this part uses double-precision variables for computation, but the tendency is saved as single precision. Gray shading indicates that this variable is diagnosed mostly from single-precision variables. Specifically, $\zeta_p = \frac{\zeta_a}{\delta\pi}$ is highly sensitive to the precision of $\delta\pi$, requiring a double precision $\delta\pi$.

For the dycore, the precision sensitivity varies among different terms. The precision-sensitive terms are primarily related to pressure gradient and gravity terms. The precision-insensitive terms are mainly advective, which may tolerate lower numerical precision. Computationally, the advective parts of the equations are using higher-order operators which are responsible for the major computational burden. The passive tracer transport equation (Eq. 10) can be mostly computed using single precision. The only part needs a careful modification is the black solid box, which indicates that it uses single-

precision variables for computing, but the result is saved as double precision. $\delta\pi V$ (representing the mass flux) in Eq. (10) is accumulated and averaged from $\delta\pi V$ in Eq. (3), so computing it uses single precision. But when using it for tracer transport, this variable is converted to double precision so that the mass continuity equation of tracer transport uses a double-precision mass flux.

The mass continuity equation Eq. (3) is solved using a flux form, ensuring global mass conservation of $\pi_s - \pi_t$ ($\pi_t$ is a constant) within the bounds of machine rounding errors, which is at the double-precision level. Using single precision $\delta\pi V$ implies that mass continuity is locally conserved at the single precision level. Recognizing the potential importance of local mass conservation (e.g., Thuburn 2008), a compilation switch is designed, so that approximating $\delta\pi V$ and the related mass continuity tendency can be achieved in either single-precision or double-precision. The time difference between approximating the continuity equation using single-precision and double-precision accounts for ~1%-2% of the total computational time. We will examine the model sensitivity to this operation in Section 4.4.

## 2.4 Model code modification

Figure 1 illustrates the modification made to the original model code repository. Thanks to the modular structure of the model code, the mixed-precision version of the model dynamics can be seamlessly integrated as an add-on component, allowing for independent development. The switch between double-precision and mixed-precision dynamics is governed by the model's control unit, facilitating the transition between two code repositories via a compiler option ("MIXCODE"). Additional adjustments for each component include modifying the parallel exchange functions to support reduced-precision variables, altering the precision level of allocated dynamics data, accommodating precision changes of specific variables in physics-dynamics coupling, and introducing a precision control variable. All these supplementary modifications are also designated by the compiler option "MIXCODE". The pure single-precision code is achieved by simply using single-precision for all variables, marked as "SPCODE" in the code. When 'MIXCODE' is defined, additional variable allocations and assignment statements are introduced. It has been confirmed that overheads due to these additional statements can be omitted, by comparing the original code and the MIX code executed in a pure double-precision mode.

## 3 Computational performance

We first examine the computational performance of the mixed-precision dynamical core in a standard reference computational test. Here, all computing tasks are carried out on a local supercomputing cluster. Each computing node is equipped with 128GB memory, and the Central Processing Unit (CPU) is a Hygon C86 7285 model at 2.0 GHz. Each CPU features a 32 KB L1 data cache, a 64 KB L1 instruction cache, a 512 KB L2 cache, and an 8192 KB L3 cache. We use "SGL" to denote pure single precision computing, "DBL" to denote pure double precision, and "MIX" to represent mixed precision computing. All experiments were conducted on a G8 grid, submitted with the same topology: 756 MPI tasks distributed across six nodes.

Compared to the double-precision model, the runtime of the mixed-precision model for the non-hydrostatic dry dynamical core (NDC), hydrostatic dry dynamical core (HDC) and tracer transport solver reduced by 27%, 24% and 44%

(Table 2). The runtime of the mixed-precision dycore solver is still larger compared to the single-precision dycore, indicating the time overhead incurred by the use of double-precision in precision-sensitive algorithms. The runtime of the mixed-precision tracer transport solver is comparable to that of the single-precision tracer transport solver, because most computations in the tracer transport module now use single-precision computing. It should be noted that the time gains from mixed-precision computing may also depend on hardware and compiler options (e.g., Brogi et al. 2024).

For real-world applications with routine I/O, the mixed-precision code maximizes its potential in the global storm-resolving model (GSRM) simulations. In Section 4.5, the MIX run achieved a 25.5% reduction in the wall-clock time for the dynamics and physics procedures (including physics-dynamics coupling), as compared with the DBL run. The simulations (5 km; 23,592,962 cells) were conducted using 3248 MPI processes distributed across 58 computing nodes, where each node is equipped with 56 Intel Xeon Gold 6348 CPUs operating at 2.60 GHz and 256 GB of memory. For both DBL and MIX runs,

the dynamics and physics procedures (including physics-dynamics coupling) accounted for approximately 95% of total wall-clock time, with dynamics alone occupying a substantial portion ranging from 83% to 85%.

For a computational task that is not significantly restricted by the memory bandwidth, the reduction of wall-clock time can be less significant. This is the case of Section 4.4, in which a coarse-resolution (120 km; 40,962 cells) model is executed using 640 MPI processes across 20 nodes. The MIX test is faster than the DBL test by roughly 12%.

As emphasized by one reviewer, reduced-precision computing can be particularly beneficial on the machines with sub-optimal interconnect, and on the Graphic Processing Unit (GPU)-like architectures, where increased computational intensity (in terms of degrees-of-freedom per GPU) can increase the overall performance. Another application of this mixed-precision code also confirms this assertion. Thanks to the Sunway's local engineers, this mixed-precision code has been successfully ported to the new Sunway supercomputer. Here, we report some observations, the detailed results will be presented

elsewhere.

One processor of Sunway has 390 cores, distributed across 6 core groups (CGs). Each CG consists of one management processing element (MPE) and 64 computing processing elements (CPEs) organized as an 8×8 array. Numerical tests were conducted at the 3.75 km and 1.875 km (icosahedral grid level 11 and 12) horizontal resolutions using the full model. A notable observation was that mixed precision typically did not yield significant speedup on the MPE side but provided

notable speedup on the CPE-parallelized kernels. Considering that the Sunway architecture generally does not exhibit higher calculation performance for single precision compared to double precision (except for division and elemental functions), we may infer that the MPE-side code is not limited by the memory bandwidth. On CPEs, the mixed-precision code demonstrates better speedup. This implies that the performance of the CPE-side code is more constrained by the memory bandwidth, and thus mixed precision computing leads to better improvements.

**4 Physical Performance**

**4.1 Moist baroclinic wave**

To ensure robustness, a hierarchy of five test cases from simple to complex is adopted for model evaluation. This first case is from the DCMIP2016, as outlined by Ullrich et al. (2014), a modified approach to the dry baroclinic instability scenario (Jablonowski and Williamson 2006). This experimental setup triggers the emergence of an unstable baroclinic wave pattern, initiated by early perturbations, which exhibits exponential growth and attains its maximum intensity around the 11[th] day. The experiment incorporates a passive tracer representing water vapor, which is subject to passive advection. Although the mixing ratio marginally influences the pressure gradient force, as noted by Zhang et al. (2020), the overall behaviour of wave growth is in substantial agreement with that in the dycore (Zhang et al. 2019). The primary objective is to assess the model's efficacy in replicating the typical dynamics of moist atmospheric conditions across various precision settings.

Figure 2 shows surface pressure and relative vorticity field at the model level near 850hPa (model layer 23th, 30 layers in total) at day 11, as simulated by the G8 resolutions. The baroclinic waves shown the anticipated growth in the DBL simulation (Figs. 2a). In the SGL simulation, the primary growth fluctuations in the DBL simulation were reproduced (Figs. 2c). However, in the Northern Hemisphere, there were developments of incorrect spurious waves, whose intensity was comparable to the major fluctuations (Figs. 2c). The Southern Hemisphere exhibited a weaker structure of spurious waves (Figs. 2c). The results from the MIX simulation displayed patterns much closer to those in the DBL simulation (Figs. 2e).

The primary difference between MIX and DBL simulations lies in the vicinity of strong gradients along the cold front (Figs. 2c). But the primary fluctuations in both MIX and DBL simulations exhibit a high degree of similarity in their patterns (Figs. 2a and 2e), indicating that precision levels have a tangible impact on the phase speed of wave propagation.

The error introduced by SGL and MIX can be quantified by comparing their solutions to a DBL solution. Following Jablonowski and Williamson (2006), $l_2$ error norms (defined in the appendix) of the relativity vorticity field at the model layer 23 are compared on the global grid as a function of time. Fig. 3 shows the $l_2$ norm for the SGL and MIX. In the initial stages of the model integration, the errors in the SGL simulations increased rapidly. By checking the original fields (figure not shown), it was found that numerous small-scale spurious fluctuations had emerged on both sides of the equator, the intensity of which was similar to the physically meaningful fluctuations.

After day 6, the primary fluctuations of the baroclinic waves in the SGL simulations began to develop, resembling the behaviour of the DBL simulations, and the errors started to decrease (Fig. 3). By day 10, the fluctuations rapidly developed, the primary fluctuations grew robustly, and the spurious fluctuations produced in the early stages of the SGL simulations also rapidly developed, leading to an increase in errors (Fig. 3). On day 11, the intensity of the spurious fluctuations developed in SGL was close to that of the primary fluctuations, which is unacceptable. Due to the slow growth of the primary fluctuations in the early stages, the MIX simulation exhibited minimal errors before day 9 (Fig. 3). Subsequently, as the fluctuations matured rapidly, larger differences in the phase speed compared to the DBL emerged, leading to a rapid increase in errors.

## 4.2 Splitting supercell thunderstorms

The splitting supercell test of DCMIP2016 (Klemp et al. 2015; Zarzycki et al. 2019) emphasizes the importance of scrutinizing non-hydrostatic model simulations of small-scale dynamics, especially as models approach spatial resolutions on

the (sub) kilometre scale. This test utilized the small-planet testing framework (Wedi and Smolarkiewicz 2009), a cost-effective approach by scaling down Earth's radius by a factor of 120. The model employs the Kessler warm-rain microphysics scheme for simplified physics. This particular test case is characterized by unstable atmospheric conditions conducive to moist convection, posing a challenge to numerical accuracy and stability. Klemp et al. (2015) suggested that an increase in horizontal resolution should lead to convergent solutions. For GRIST, this behaviour has been verified by Zhang et al. (2020). Our investigation further examines the capability of the MIX configuration to accurately replicate the behaviours observed in the DBL simulations.

Figure 4 shows the $q_r$ mixing ratio at 5 km elevation in both DBL and MIX simulations at four resolution choices (G4: ~4 km, G6: ~1 km, G7: ~0.5 km and G8: ~0.25 km). The DBL and MIX solutions show bulk similarities across all the resolutions. At 7200 s, a single updraft splits and evolves into a symmetric storm propagating towards the poles, with two supercells located ~30° from the equator. These supercells show subtle differences in their structure and intensity. At a low resolution of 4 km, the differences between MIX and DBL simulations are minimal at all altitudes (Figs. 4b-c). As the resolution increases from 4 km to 1 km and to 0.5 km, the structural differences in supercells gradually become more pronounced (Figs. 4b-d, f-h, j-l). However, when the resolution further increases from 0.5 km to 0.25 km, the differences diminish (Figs. 4n-p). For DBL simulation results, the differences between 0.5 km and 0.25 km are smaller than those between 1 km and 0.5 km, indicating that the solution converges almost at a resolution of 0.5 km. At 0.25 km, the results of MIX simulation show greater similarity to those of DBL simulation at all altitudes (Figs. 4n-p). This indicates that, in the mixed-precision simulation, supercells also achieved good convergence at this resolution, and thus the sensitivity to the precision level diminishes from 0.5 km to 0.25 km.

Figure 5 shows the maximum vertical speed and area-integrated rainfall rate over the global domain as a function of time for each resolution. The vertical speed in both MIX and DBL increases with resolution (Fig. 5a). From the start of the model integration until 5400s, the vertical speed curves of MIX and DBL simulations nearly overlap (Fig. 5a). After 5400s, a noticeable deviation appears, excepted for the G4 grid. The difference in vertical speed between MIX and DBL is minimal at 4 km resolution, followed by the 0.25 km resolution, while it is larger at 1 km and 0.5 km resolutions (Fig. 5a). The area-integrated rainfall rate curves exhibit similar evolutionary features (Fig. 5b). At very low resolutions, such as 4 km, the differences between MIX and DBL simulations are not significant. At a higher resolution of 0.25 km, the overall behaviour of supercells in the MIX simulations is closer to that of DBL compared to 0.5 km and 1 km resolutions. Both MIX and DBL solutions exhibit convergence behaviours.

**4.3 Idealized tropical cyclone**

This idealized tropical cyclone scenario integrates a three-dimensional dynamical core with a simple physics suite (Reed and Jablonowski 2012), alongside an analytic vortex initialization technique (Reed and Jablonowski 2011). The experiment produces the evolution of a tropical cyclone from a nascent, idealized vortex, highlighting the model's sensitivity to various parameter adjustments. Notably, alterations in the tracer transport schemes of GRIST can produce subtle

sensitivities in the development of the tropical cyclone due to the pressure gradient terms (Zhang et al. 2020), thereby
establishing this case useful for assessing model precision sensitivity.

Figure 6 displays the wind speed at day 10 for the DBL (Figs. 6a and 6b) and MIX (Figs. 6c and 6d) simulations at the G8 resolution. Fig. 6 (left) shows the longitude-height cross sections of the magnitude of the wind through the centre latitude of the vortex. Fig. 6 (right) displays the horizontal cross sections of the magnitude of the wind at the lowest model layer. The centre of vortex is defined as the grid point with the minimum surface pressure. At the day 10, the developed storm resembled a tropical cyclone. The overall behaviour in the MIX simulation was similar to that in the DBL simulation, with maximum winds near the surface and a distinct eyewall structure (Fig. 6). However, there was some differences in the vertical structure and centre location of the cyclone (Figs. 6a and 6c). In the MIX simulation, the generated cyclone was stronger, with higher wind speeds near the surface (Fig. 6c). The eyewall of the cyclone in the MIX simulation appeared less pronounced compared to that in the DBL simulation, where the cyclone's eyewall is narrower and straighter (Fig. 6c). Overall, the characteristics of the cyclone were comparable between the MIX and DBL simulations.

In addition to two deterministic control simulations using both double-precision and mixed-precision with the non-hydrostatic solver, eight ensemble simulations with the double-precision non-hydrostatic model are further performed. This assesses the MIX simulation within the uncertainty range of the DBL simulation. The uncertainty range is quantified by the ensemble simulations encompassing eight initial-value perturbation members. Random small-amplitude perturbations were applied to the initial wind speeds (e.g., Li et al. 2020), where perturbations to the normal velocity at cell edges were prescribed within a range of 2% of their values in the control experiment.

Figure 7 describes the tracks of tropical cyclones, along with the evolution of minimum surface pressure and maximum surface wind speed over time. The red and blue lines represent two deterministic simulations conducted using MIX and DBL solvers, respectively. The eight random perturbation simulations with the DBL solver are represented by gray lines. Minimal spread is observed in the early stages of the simulations (Fig. 7). Cyclone track separation between MIX and DBL simulations occurs at the day 1 (Fig. 7a). Subsequently, spread in the simulations increases over time (Fig. 7). The evolution of minimum surface pressure and maximum surface wind speed over time exhibits similar trends (Figs. 7b,7c). No discernible difference is found between the sensitivity introduced by MIX and that introduced by perturbed DBL simulations. The overall behaviour of the MIX simulation falls within the range of uncertainty of the DBL simulation.

## 4.4 Atmospheric Model Intercomparison Project (AMIP) simulation

Following the establishment of the MIX dynamical core, a detailed examination of its integration with the model physics suite (Li et al. 2023) becomes crucial. The nonlinear interactions between the model's dynamics and its physical processes can result in varied performances across weather and climate simulations. It's imperative to investigate these differences to ensure that MIX simulations can accurately mirror the outcomes of DBL simulations in practical applications.

In assessing a new formulation for real-world modelling, our guiding principle is to first run long-term AMIP simulations (Zhang et al. 2021). This ensures that the model can achieve statistical equilibrium, maintain a realistic model climate, and has good integral properties such as conservation and balanced budgets (e.g., Fu et al. 2024). Subsequently, the

same model, with minimal application-specific modifications, undergoes shorter-range but higher-resolution, kilometre-scale tests (Zhang et al. 2022).

The AMIP experiment is conducted in alignment with Zhang et al. (2021). This involved running both hydrostatic and non-hydrostatic models with the weather physics suite at a G6 grid over a decade, spanning from 2001 to 2010. The simulations were performed under the conditions with prescribed climatological sea surface temperatures and sea ice concentrations. The focus was narrowed to precipitation, which is a comprehensive metric due to its sensitivity to both model dynamics and physics, effectively reflecting the non-linear interactions that are crucial for accurate weather and

climate simulations (Zhang and Chen 2016).

Figure 8 shows the simulated climatological (2001-2010) precipitation field for June-July-August (JJA) and December-January-February (DJF). Both the MIX hydrostatic and non-hydrostatic solvers can replicate the JJA and DJF precipitation patterns in the DBL simulations. The discrepancies between MIX and DBL simulations are similar in both hydrostatic and non-hydrostatic simulations, with the primary differences occurring in the tropics. The precipitation differences shift from

north to south along with the main rain bands as the season transition from (boreal) summer to winter. The deviation in summer precipitation is greater than that in winter precipitation, because convective activities are most vigorous. In the summer, the MIX simulation overestimates the precipitation in the tropical coastal regions of the Western Pacific, especially along the western coast (Figs. 8a and 8b). In winter, the main biases in the MIX simulation are concentrated in the Southern Ocean (Figs. 8c and 8d).

These results may have two implications. In MIX simulations, the cumulative effects of rounding errors might be progressively magnified over the course of long-term climate integrations. This phenomenon could lead to notable differences in the simulated large-scale atmospheric phenomena. This contrasts with high-resolution shorter-range weather modeling, where discrepancies primarily emerge at the small scales, as will be discussed in Section 4.5. This might imply that MIX may diverge more from their DBL counterparts over extended integration periods, necessitating a careful

consideration of how rounding errors accumulate and their impact on the climate simulation performance.

Second, the differences induced by varying the precision level can be further exacerbated by physical processes within the climate system. A clear example is observed in the tropical regions during the boreal summer, where higher discrepancies are noted. This suggests that certain atmospheric conditions or regions, such as the tropics during periods of intense solar heating, may be more susceptible to the effects of precision-level differences. These conditions can amplify the

inherent precision differences, leading to more pronounced variations.

In the MIX implementation, Eq. 3 implies that global mass is conserved at the double precision level. The local mass flux is only conserved at the single precision level, because the mass flux and its divergence are treated as single precision. As mentioned in Section 2.3, we have retained a capability to compute the terms related to the mass flux divergence equation also in the double precision. Local mass can be conserved at the double precision level as well. We then evaluated the long-

term climate integration results based on the hydrostatic model.

Figure 9 shows the differences of the climatological precipitation field between MIX with single- (MIX_SGL_mass) and double-precision (MIX_DBL_mass) mass flux divergence against the pure DBL simulation. In the summer, the simulation differences between the MIX_SGL_mass and MIX_DBL_mass solver are small (Figs. 9a and 9b). In the winter, the deviations in the MIX_SGL_mass is smaller than those in the MIX_DBL_mass solver (Figs. 9c and 9d). The deviations are most pronounced in the tropical convective precipitation over the southern tropical oceans (Figs. 9c and 9d). The larger difference between MIX_DBL_mass and DBL is *likely* due to implicit type conversions, as discussed in Section 2.2.

## 4.5 A global storm-resolving simulation

Under the constraints of today's computational resources, executing GSRM nonhydrostatic simulations remains resource intensive (Satoh et al. 2017; Stevens et al. 2019). The use of MIX simulations presents a cost-effective solution to this challenge. However, it has been reported, for instance by Nakano et al. (2018), that as the resolution of the model increases, the difference between MIX and DBL may increase, especially for the smaller-scale flow features. This observation prompts a closer investigation into the performance of nonhydrostatic models at high-resolution modelling.

A GSRM experiment at 5 km (G9B3) is performed using the MIX nonhydrostatic model, following Zhang et al. (2022). The model was integrated from 0000 UTC, July 10, to 0000 UTC, July 15, 2015. We expect that the developed mixed-precision dynamical core can replicate the behaviour of DBL in the kilometre-scale weather simulations.

Figure 10 show the period-accumulated precipitation (0000 UTC, July 10, to 0000 UTC, July 15) from the MIX and DBL model runs. All data have been interpolated to a $0.5°$ regular latitude-longitude grid. The precipitation pattern simulated by MIX are very close to those of DBL simulations. MIX obtains nearly the same general position, orientation, and intensity of the rain band (Figs. 10a, 10b). MIX and DBL also produced very comparable kinetic energy spectra (figure not shown).

Like the AMIP simulations, the differences in precipitation are primarily located within the tropics, with the most pronounced differences in areas with vigorous convection. Close-ups of these locations reveal that it is small-scale (a few grid spaces) that is most sensitive to the precision level, because small scales are most sensitive to numerical discretization and dissipation (Jablonowski and Williamson 2011). Considering that global meso-scale forecast at a few kilometres would greatly benefit from ensemble prediction (Palmer 2019), in practice, the MIX induced small-scale sensitivity may also fall within the uncertainty range of the ensemble, similar to that in Section 4.3.

## 5 Discussion

As mentioned in Section 2.3, the advective parts of the equations are not sensitive to the precision level and they are using higher-order operators. To understand whether this optimization outcome is sensitive to the nominal order of numerics, we utilized an isolated 3D tracer transport experiment (Kent et al. 2013, Hadley-like meridional circulation) and performed a convergence test. This test case was also performed by Zhang et al. (2020) and Li and Zhang (2022), and thus their results can be used as reference. By adjusting the very small values introduced by the limiter to be within the single-precision range, this equation (Eq. 10 in Fig.1) is solved independently in single-precision ("SPCODE" compiler option in the model code).

We used various orders of horizontal flux operators, namely RK3O2, RK3O3, and RK3O4 (combinations of a third order Runge-Kutta integration scheme and the nominal 2nd- to 4th-order spatial flux operators). RK3O3 is used in other tests of this paper. The vertical advection operator remains unchanged. The results are shown in Fig. 11. The tested resolutions and associated time steps include G5L30 (600s), G6L60 (300s), G7L120 (150s), G8L240 (75s) and G9L480 (37.5s). The results demonstrate that, using different-order horizontal flux operators, the single-precision simulations are comparable to the double-precision simulations across all resolutions, with nearly identical error norms and convergence rates.

This outcome suggests that, within the current code implementation, the advective part of the model demonstrates greater resilience when subjected to changes in precision, regardless of the nominal order of the numerical operator. This supports the optimization results in the entire dynamical equations. As reported by Nakano et al. (2018) and Yin et al. (2020), the precision-sensitive components are related to the specific numerical algorithms. Meanwhile, badly conditioned code or poor coding practice may also necessitate double-precision calculations (Váňa et al. 2016; Palmer 2020). For other components (e.g., pressure gradient) that currently have higher sensitivity to the precision level, we believe that it may require more careful code implementation to allow us to benefit more from reduced-precision computing.

We consider this project to be a success at least in its current phase. Existing literature and our own experiments suggest that the deviation between reduced-precision and double-precision codes tends to amplify with higher resolutions. Therefore, the mixed-precision code optimization developed based on the G8-grid test can have relatively smaller deviations from the double-precision model on the coarser grids. While we cannot 100% guarantee that the optimization outcome is optimal for all the grid resolutions, the current 5 km test is also reasonable. We are overall confident that the present code will not degrade the operational skill score (e.g., those examined by Wang et al. 2024), but more testing efforts are still required for quality operational runs. In the future, we aim to further reduce the precision of certain variables and conduct more tests at the kilometre-scale to ensure the robustness of the optimized code. Some alternative advection schemes in the tracer transport module have not been implemented to single precision yet and this can be done in future. Experiments with further reduced significant digits also deserve exploration.

## 6 Summary

In this study, we investigated mixed-precision computing within the GRIST dynamical core, identifying the equation terms particularly sensitive to numerical precision. We outlined an optimization procedure characterized by a limited extent of iterative development. Given the current development trajectory of high-performance computing, where advancements in memory bandwidth lag behind peak processor performance improvements, mixed-precision computation holds promise for enhancing weather and climate model development. The major conclusions are summarized as follows.

We discovered that terms sensitive to numerical precision primarily involve pressure gradient and gravity. In contrast, advective terms exhibit resilience to single precision and can be optimized. The advective terms are computationally more expensive than the pressure gradient and gravity terms. The viability of employing mixed-precision computing in the GRIST dynamical core has been validated across a spectrum of scenarios, from idealized flow to real-world AMIP and GSRM

simulations. These MIX experiments yielded results remarkably similar to those from the DBL simulations. For dycore, the runtime for the dry hydrostatic and dry non-hydrostatic cores was reduced by 24% and 27%, respectively. The tracer transport module witnessed a runtime reduction of 44%. The overall time savings depend on the proportion of dycore and tracer transport in the total wall-clock time, and the scale of a computational task, varying by application. For instance, the MIX-GSRM experiment in Section 4.5 witnessed a 25.5% reduction in the wall-clock time, as compared with the DBL-GSRM experiment.

We noted a higher sensitivity to precision in long-term climate simulations compared to short-term higher-resolution weather simulations, particularly affecting the precipitation field over certain regions. In shorter-range weather forecast, the differences between MIX and DBL are mainly found for the small scales, while in the AMIP simulations, the difference is found for the larger scales. These effects may primarily stem from the model sensitivity to the precision level or from biases introduced by mixed-precision computations themselves. Compared with the low-resolution global simulations, the mixed-precision code is more beneficial for the GSRM simulations at a few kilometres, or other model applications with comparable computational scales as GSRMs.

**Acknowledgments**

Editors and reviewers are thanked for their handling and comments of this paper. This study is supported by a National Talent Project (2021).

**Code and Data availability**

Model code and plotting data related to this manuscript is available at: https://zenodo.org/records/11229770.

**Author contribution**

SYC developed the mixed precision model code and prepared the initial draft. YZ designed and led this model development research. YW contributed to experiments. All the authors discussed this work and contributed to the final manuscript version.

**Competing interests**

None.

**Appendix**

We define the three-dimensional global integral of $\mathcal{H}$ as:

$$I(\mathcal{H}) = \int_{z=z_{surface}}^{z=z_{top}} \oiint \mathcal{H} \, dAdz, \quad (A1)$$

where $A$ denotes cell area and $z$ denotes height. The vertical integral is omitted if two-dimensional space is under consideration. The definitions of $L_1$, $L_2$, and $L_\infty$ are as follows:

$$L_1 = \frac{I(|\mathcal{H}-\mathcal{H}_T|)}{I(|\mathcal{H}_T|)}, \quad (A2)$$

$$L_2 = \sqrt{\frac{I[(\mathcal{H}-\mathcal{H}_T)^2]}{I[(\mathcal{H}_T)^2]}}, \quad (A3)$$

$$L_\infty = \frac{max\forall|\mathcal{H}-\mathcal{H}_T|}{max\forall|\mathcal{H}_T|}, \qquad (A4)$$

where $\mathcal{H}$ and $\mathcal{H}_T$ are the computational solution and true solution, respectively. Max $\forall$ means selecting the maximum value from the field.

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

**Table 1.** Grid Name and Corresponding Horizontal Resolutions.

| Grid level of a subdivided icosahedron | Horizontal resolution (km) on a full-size Earth and on a small-radius Sphere (if used) | Number of Cells |
|---|---|---|
| G4 | 480 km (4 km) | 2,562 |
| G5 | 240 km | 10,242 |
| G6 | 120 km (1 km) | 40,962 |
| G7 | 60 km (0.5 km) | 163,842 |
| G8 | 30 km (0.25 km) | 655,362 |
| G9 | 15 km | 2,621,442 |
| G9B3 | 5 km | 23,592,962 |

**Table 2.** Elapsed time using single-, mixed- and double precision **(**The runtime of each solver is normalized to that of the corresponding solver in double-precision**).**

| Grid Name | Precision | Dycore time (1440 steps) | | Tracer time (1440 steps) |
|-----------|-----------|-----------|-----------|-----------|
| G8 | DBL | 1 (NDC) | 1 (HDC) | 1 |
| | SGL | 0.53 (NDC) | 0.56 (HDC) | 0.58 |
| | MIX | 0.73 (NDC) | 0.76 (HDC) | 0.56 |


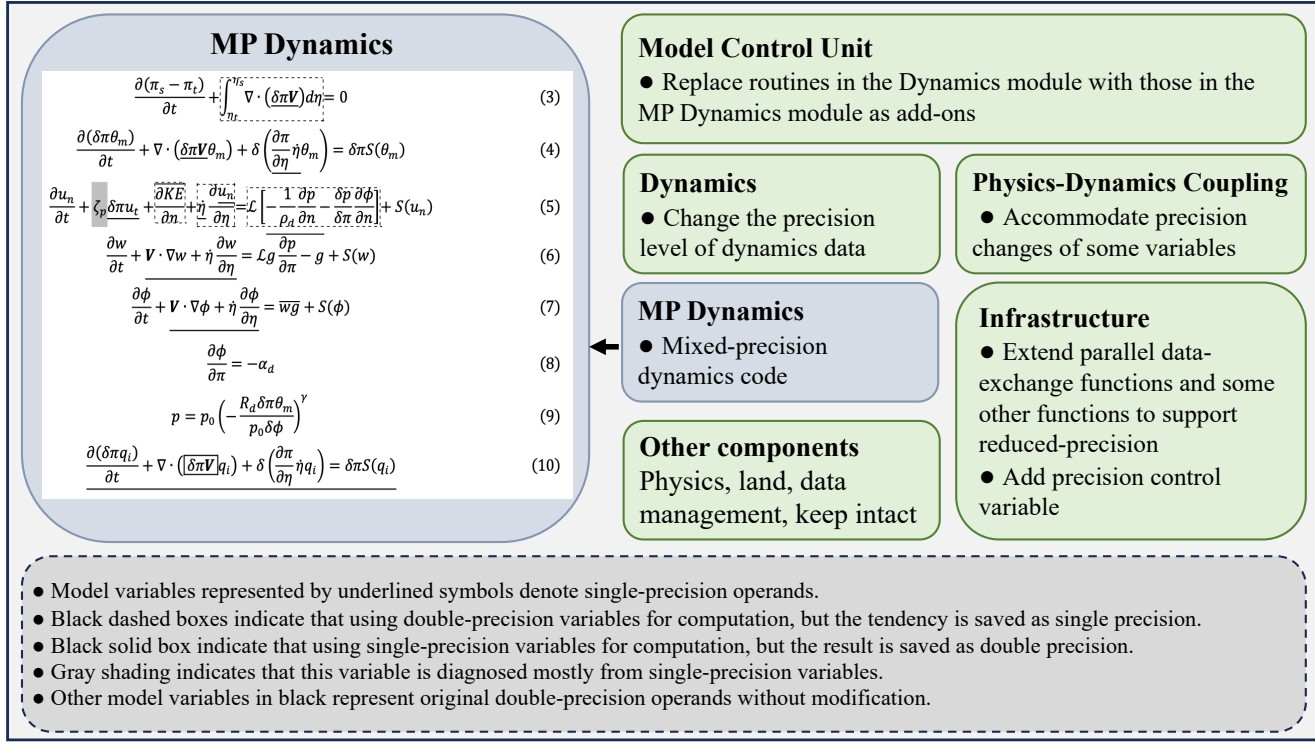

**Figure1: Modifications to the GRIST model code repository for implementing the mixed-precision dynamical core.**

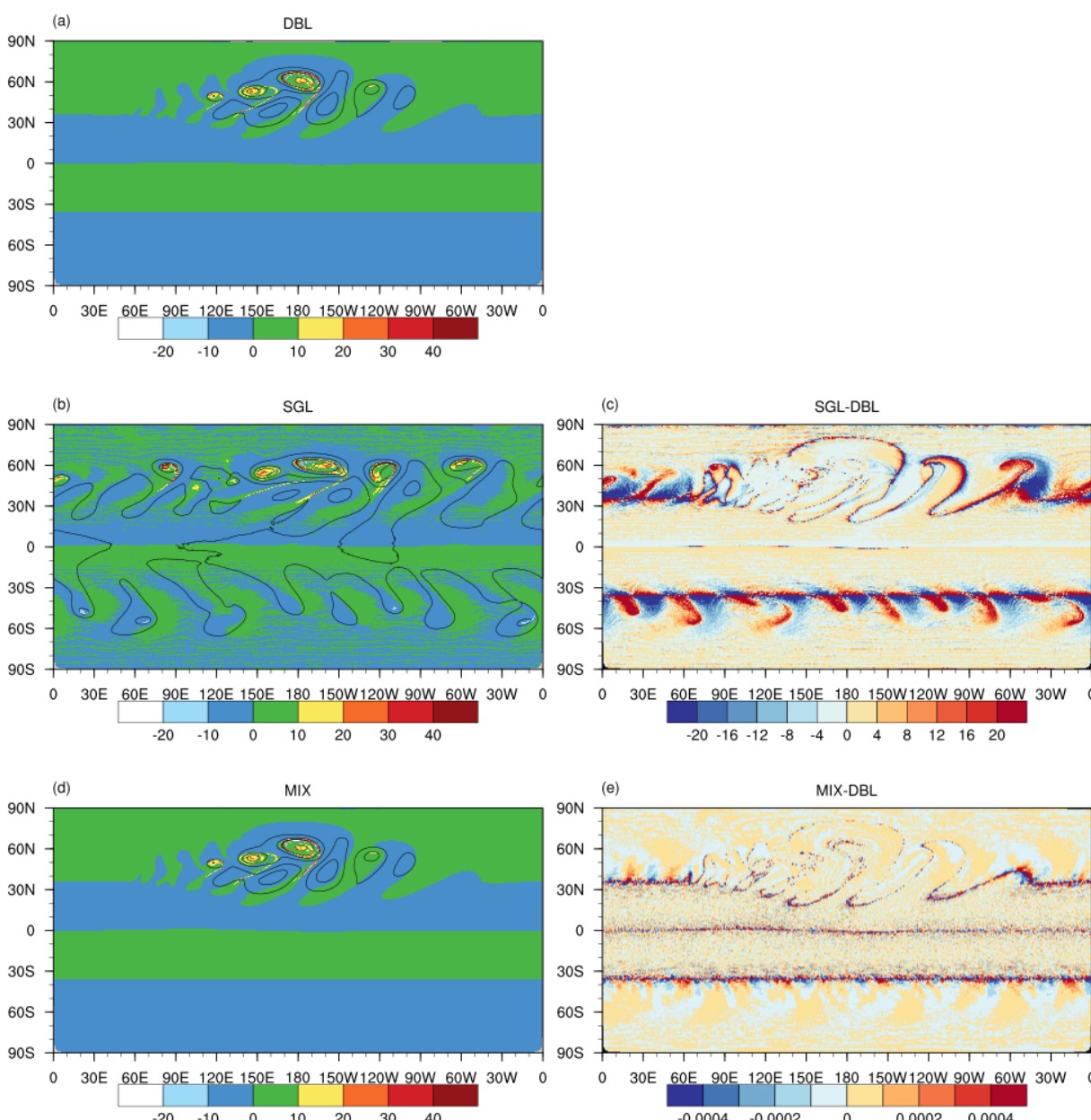

**Figure 2: Baroclinic wave development at day 11 in the (a) DBL simulation, (b) SGL simulation and (d) MIX simulation. (left) Colors show relative vorticity ($\times 10^{-5}\ s^{-1}$) and contours of the surface pressure and (right) the relative error between SGL and DBL, as well as the difference between MIX and DBL.**



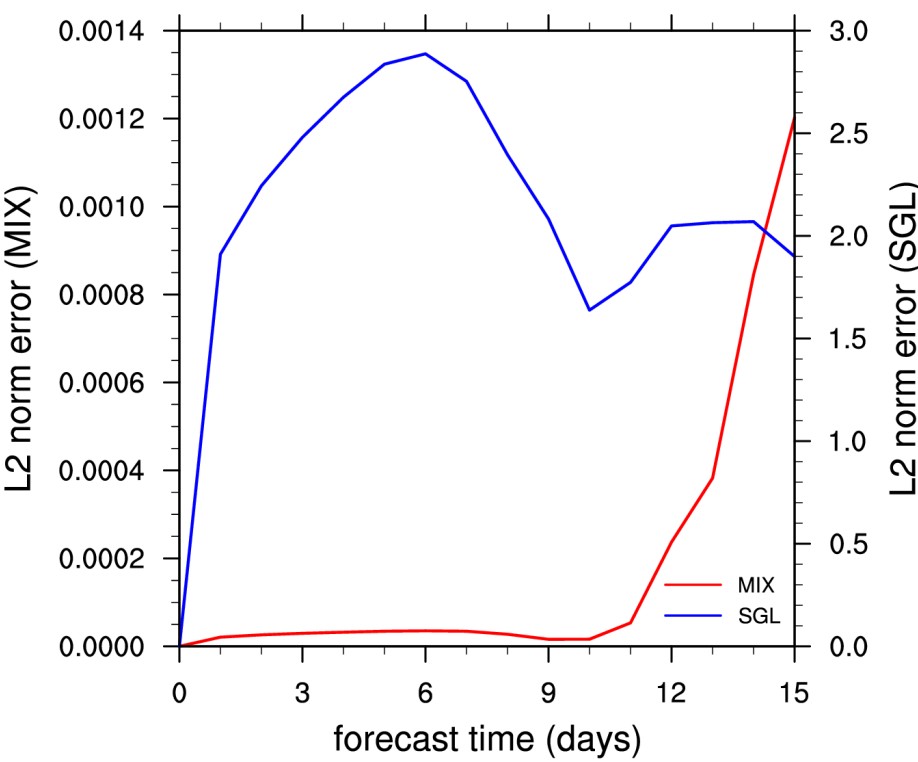

**Figure 3: Time evolution of global $l_2$ difference norm of simulated relative vorticity between the SGL and DBL, as well as $l_2$ difference norm between the MIX and DBL. Red and blue represent SGL and MIX experiments, respectively.**


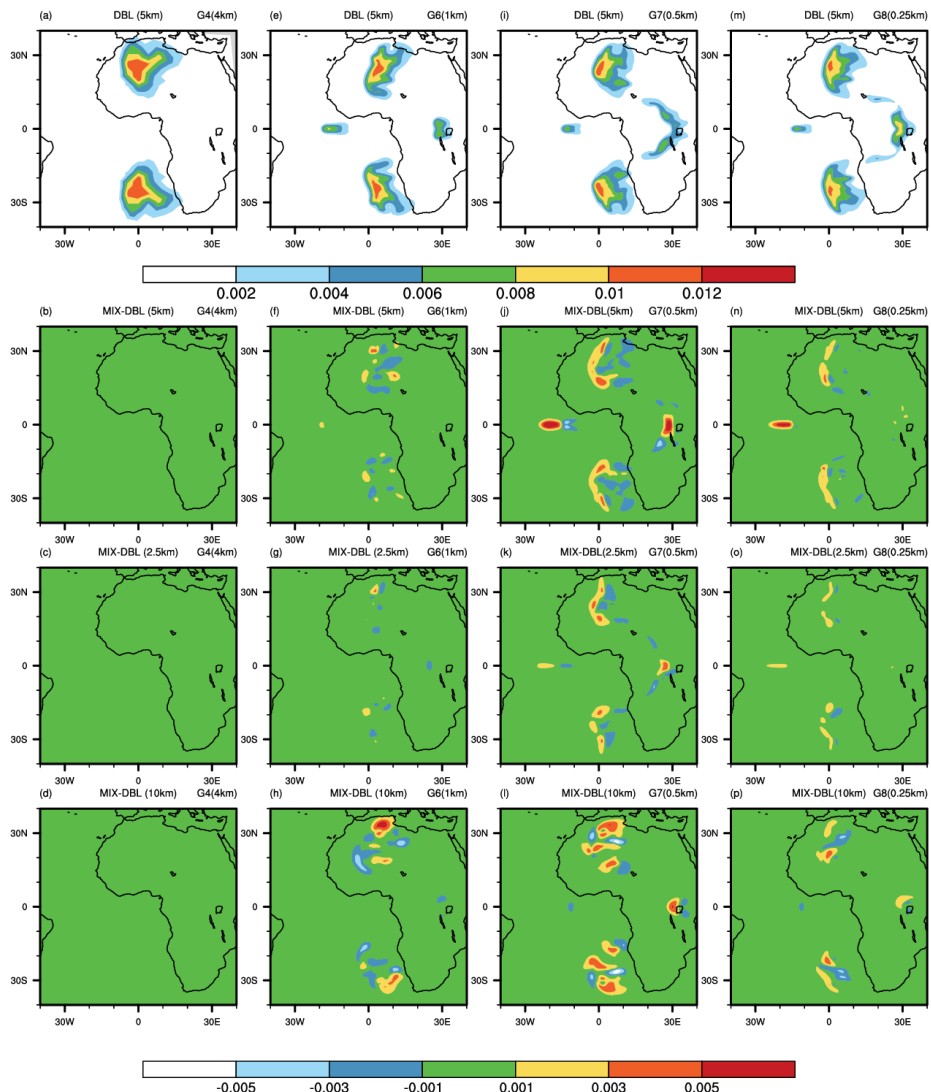

**Figure 4: Horizontal cross-sections of rainwater mixing ratio at different heights from supercell thunderstorms simulations. The first row displays double-precision simulations at 5 km altitude. The second, third, and fourth rows show the differences between mixed-precision and double-precision simulations at 5 km, 2.5 km, and 10 km altitudes, respectively. The four columns represent results at different resolutions: G4 (4 km), G6 (1 km), G7 (0.5 km), and G8 (0.25 km) from left to right.**

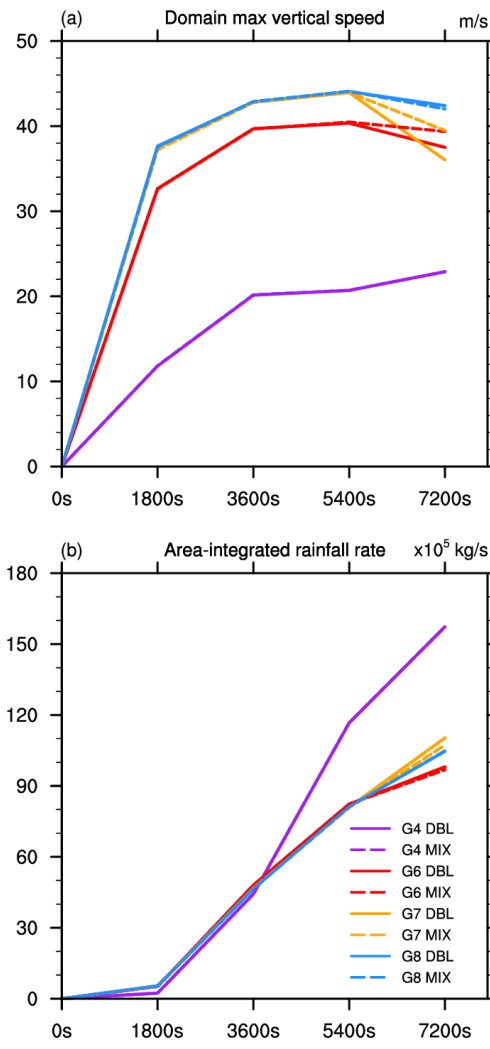


**Figure 5: The (a) domain maximum vertical speed and (b) area-integrated rainfall rate obtained from the supercell simulations.**

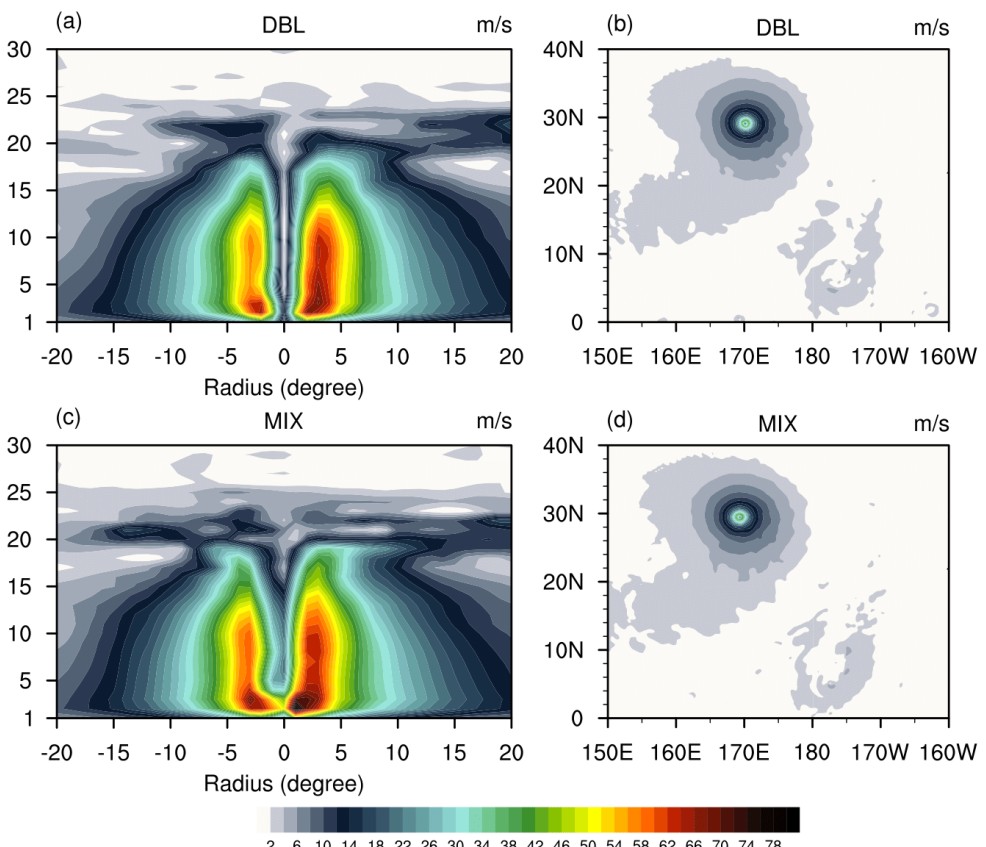

Figure 6: The simulated wind speed (m s$^{-1}$) at the G8 resolution with NDC solver, including MIX (a, b) and DBL (c, d) simulations. (left) Longitude-height cross section of the wind speed through the center latitude of the vortex as a function of the radius from the vortex center. (right) Horizontal cross section of the wind speed at the lowest model layer.

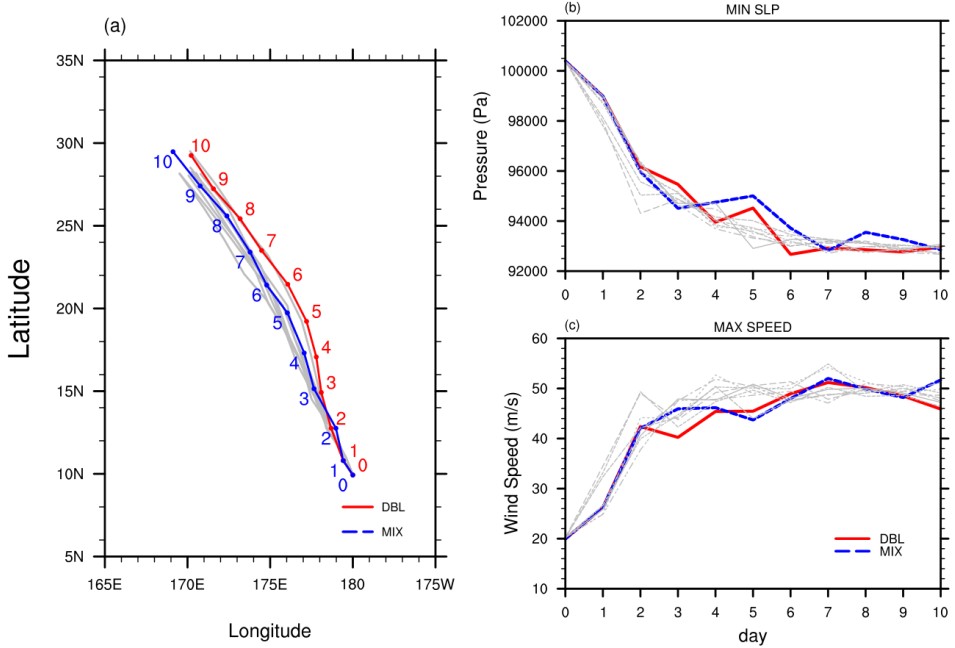


**Figure 7: The results from the deterministic and ensemble simulations. (a) The track of the tropical cyclone center for MIX (blue lines) and DBL (read lines) deterministic simulations. Time evolution of the (b) minimum surface pressure and (c) maximum surface wind speed from the and deterministic and ensemble simulations. The red and blue lines represent the deterministic MIX**

**and DBL simulations, respectively. The gray lines represent the eight runs with random perturbations to initial normal velocity at cell edges.**

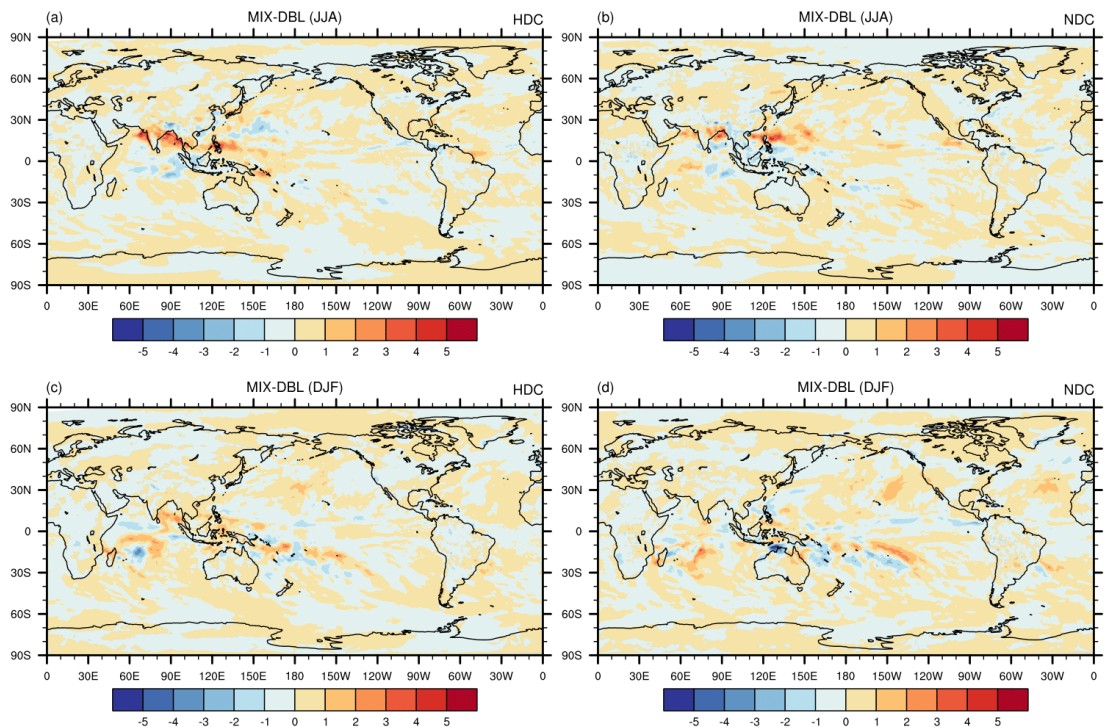

**Figure 8:** The difference between the MIX and DBL simulations, including solutions from the hydrostatic (left column) and non-hydrostatic (right column) solver. The first and second rows respectively display the averaged (2001-2010) precipitation rate (mm/day) for JJA and DJF.

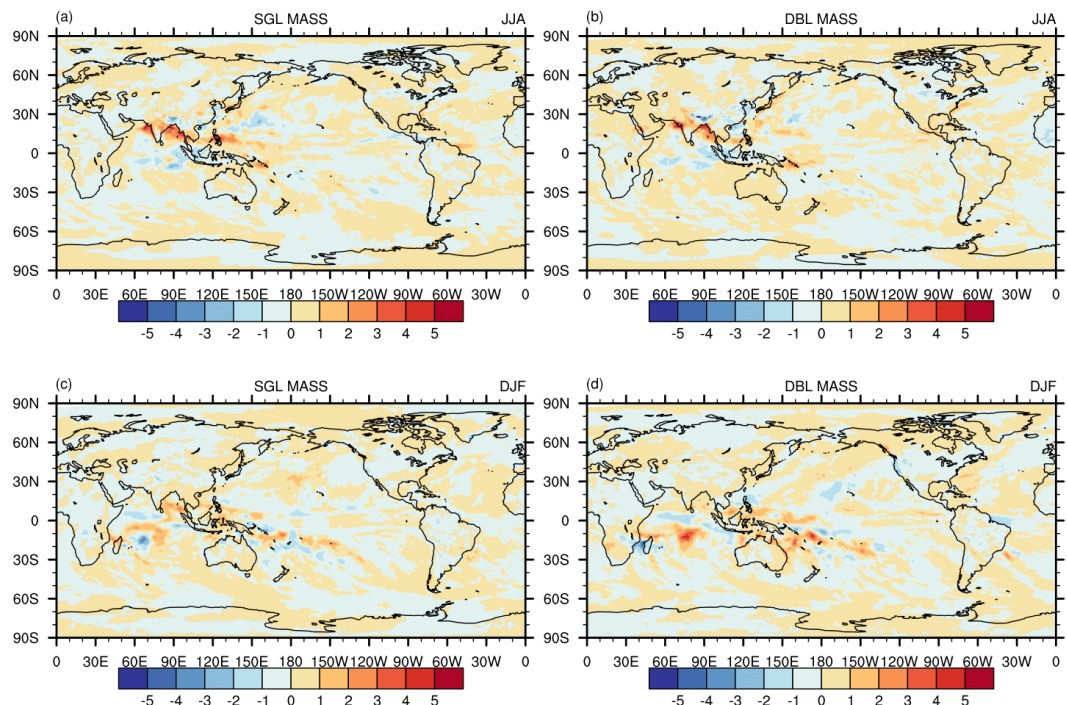

**Figure 9: (a) Difference between the JJA-averaged (2001-2010) precipitation rate (mm/day) simulated by the SGL continuity equation solver in mixed-precision mode and the "true DBL value"; (b) same as (a) but for the DBL continuity equation solver. (c)-(d) same as (a)-(b) but for the DJF-averaged (2001-2010) results.**

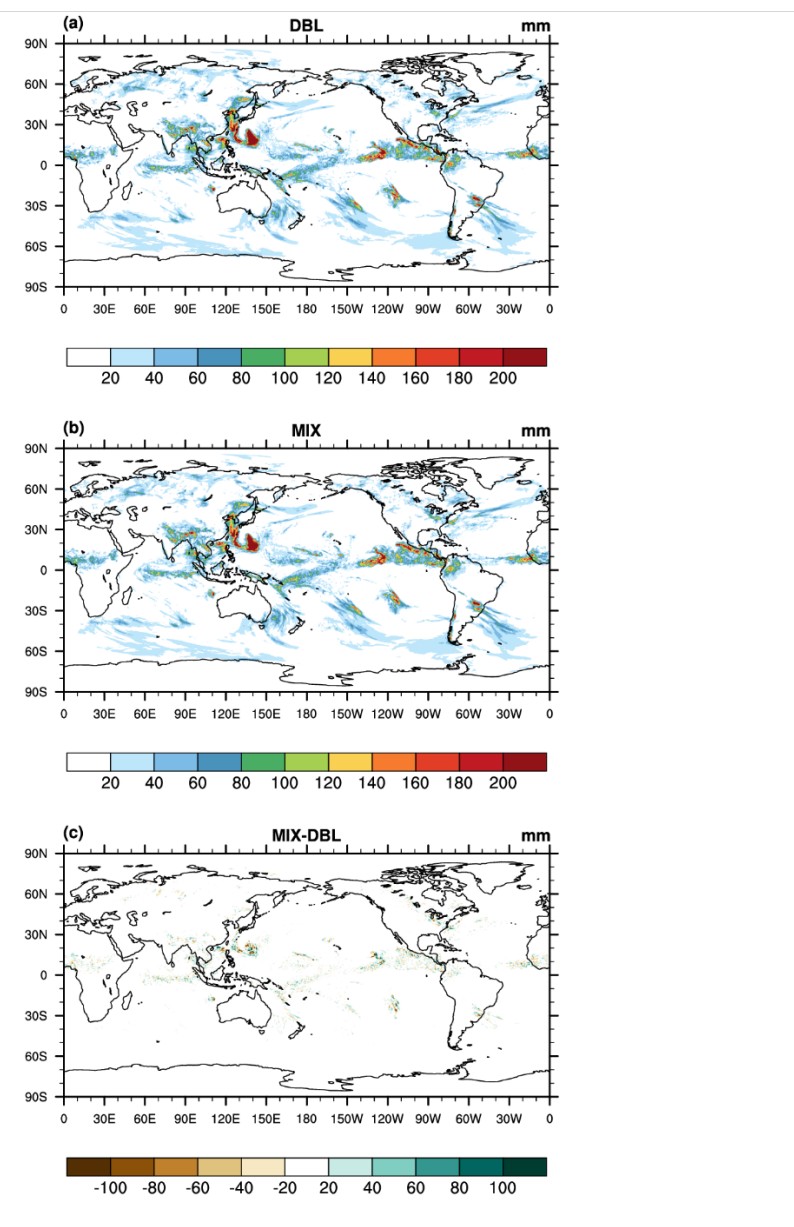

**Figure 10: The 5-day (0000 UTC, July 10, to 0000 UTC, July 15, 2015) accumulated precipitation (units: mm) from the (a) DBL simulation, (b) MIX simulation and (c) the difference between MIX and DBL simulations.**

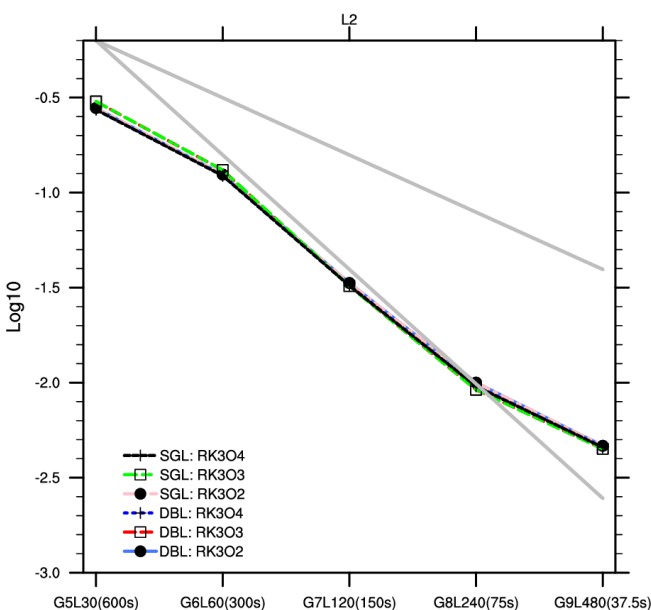

**Figure 11: The $L_2$ norm error and convergence rate of 3D passive transport test (Hadley-like meridional circulation) using single (SGL) and double (DBL) precision code for three horizontal flux operators: RK3O2, RK3O3, and RK3O4. The upper and lower grey lines correspond to the slopes of first- and second-order convergence rates, respectively.**