# Peer review of "Mixed-Precision Computing in the GRIST Dynamical Core for Weather and Climate Modelling"

_Geoscientific Model Development, 2024_

## Author Comment (AC1)

**General comments:**

The paper provides a good description of a single and mixed precision approach in atmosphere modeling, with adequate motivations for current and future models. The authors describe their strategy for building the single and mixed precision models, as well as the justification for their choices. The results are divided in a computational performance and a physical performance sections, which help the reader understanding the benefits/disadvantages of each approach both from a computational perspective as well as from a model skill one.

Overall, this is a good paper, and I recommend its publication. However, I would like to suggest a few aspects that the authors could clarify better, or provide some perspective on.

Reply:

We would like to express our gratitude to Dr. Luca Bertagna for detailed review and constructive feedback on our manuscript. We appreciate your insightful comments and suggestions. Each issue you highlighted has been thoroughly addressed in our revised manuscript. Your expertise has been essential in enhancing both the quality and clarity of our work.

**Specific comments:**

1. From the text in section 2.2, it seems the authors adopted a "greedy" approach, where each choice of variable to treat in single precision is built on top of the previous. Is that the case? Have they tried to start from different choices for the initial single precision variable? If so, what did they observe? If not, do they expect differences? I'm thinking that all paths would led to a similar final configuration of variables to be switched to single precision, but would be nice to know for sure if that's the case or if different initial choices may lead to different final configurations.

**Reply:**

Yes, a greedy-like approach was employed to identify precision-sensitive variables, where the selection of each single-precision variable builds upon the preceding one. The computational sequence of the code mirrors that of the double-precision version. Initially, in selecting single-precision variables, we systematically attempted to reduce the precision of variables encountered sequentially in the code, starting with the first variable, followed by the second, third, and so forth. We did not explore alternative initial choices for single-precision variables.

Overall, we are confident that various paths would converge to comparable final configurations of single-precision variables. As detailed in Section 2.2, the precision sensitivity varies among different computational terms, with some being highly sensitive while others are less so. This suggests that precision sensitivity is inherent to specific terms and largely unaffected by the chosen path.

Nonetheless, we acknowledge the potential for further optimization and time reduction.

2. The authors say that "the precision optimization tests were conducted using the G8 grid". I assume the resulting configuration of single precision variables was also used for the G6/G7 tests in the section 4.2. Have the authors confirmed that the same configuration was indeed optimal (by means of adding/removing some SGL

variables) also for other grids? If not, would they expect any difference? Why or why not?

**Reply:**

The initial development of the mixed-precision code was mainly based on the G8 grid and the Jablonowski-Williamson (dry) baroclinic wave test, as described in Section 2.2. Subsequently, this code was utilized for all tests presented in this paper, encompassing both higher and lower resolution grids.

While we cannot guarantee that the optimization results are universally optimal across all grids, existing literature and our own experiments suggest that the deviation between reduced-precision codes and double-precision codes typically tends to amplify with higher resolutions. Therefore, we tentatively hypothesize that the mixed-precision model derived from the G8 grid demonstrates relatively minor deviations from the double-precision model on coarser grids. Additionally, the current 5km test appears reasonable for higher-resolution grids.

In the future, we aim to further reduce the precision of certain variables and conduct more tests at the kilometer-scale to ensure the robustness of the reduced-precision model code.

3. When formatting equations in section 2.3, be aware of color-blind people. I am not one, so I can't give a thumbs up/down. But I would suggest to verify that the color choices are not affecting color-blind readers. E.g., the blue term in eq 10 may not look blue, or the green term in eq 5 may not stand out from the red around it. As a possible alternative, you could consider underlying, or use another box (perhaps dashed, to distinguish from the black one).

**Reply:**

In response, we have revised the presentation of the equations in Section 2.3, utilizing symbol annotations instead of color markings for clarity. Additionally, we have compiled the equations alongside the code modifications in an illustration (Fig. 1).

4. In section 3, the authors mention that SGL, MIX, and DBL all used the same computational resources. I would assume that, among other things, a reduced precision could allow to use less computational resources, which would additionally benefit performance (due to reduced MPI costs). This can be particularly beneficial on machines with sub-optimal interconnect, as well as on GPU architectures, where increased computational intensity (in terms of degrees-of-freedom per GPU) can increase overall performance. Have they explored this avenue? What are their thoughts on this?

**Reply:**

We agree "a reduced precision could allow to use less computational resources, which would additionally benefit performance (due to reduced MPI costs)." This is indeed the case. With fewer computational resources, the memory usage per processor increases, exacerbating memory bottlenecks. The use of mixed precision can alleviate the memory bandwidth bottleneck.

While not pursued in this paper, there is another work that has ported the mixed-precision code of this study to the next-generation Sunway computer. The processor of Sunway has 6 core groups (CGs), each of which consists of one management processing

element (MPE) and 64 computing processing elements (CPEs) organized as an 8×8 array (390 cores per processor). Those tests were conducted at 1-3 km horizontal resolutions. This may correspond to the GPU-like situation you mentioned, where "increased computational intensity can increase overall performance".

A notable observation is that mixed precision typically does not yield significant speedup on the MPE side but provides notable speedup on CPE-parallelized kernels. Considering that the Sunway architecture generally does not exhibit higher calculation performance in single precision compared to double precision, except for division and elemental functions, we can infer from the results that the MPE code is computation-bound. On CPEs, mixed-precision code demonstrates better speedup. One possible reason is that the CPE code appears to be constrained by memory bandwidth, and mixed precision reduces data size, conserving memory bandwidth and increasing cache hit ratio.

5. On line 182: why is 24% in parentheses? Seems like the line should be "27%, 24%, and 44%".

**Reply:**

Thank you. This was an error, and we have corrected it in the revised manuscript as "27%, 24%, and 44%".

6. Fig 2 seems to show absolute L2 error. It may be helpful to show a relative error, so that the reader can better gauge the impact of the SGL/MIX approximations.

**Reply:**

We apologize for not clearly defining the error norms in the initial manuscript. We have added an appendix in the revised manuscript to explain the error definition. Each error norm is now normalized by the true value of each variable, so the errors in Figure 3 (initially Figure 2 in the submitted manuscript) are relative errors.

Additionally, we have changed the errors in Figure 2 (initially Figure 1 in the submitted manuscript) from absolute to relative errors to present the results more intuitively. Correspondingly, we have revised the text to reflect this change.

7. Have the authors tried to see whether, for a fixed set of SGL variables, the quality of the SGL/MIX approximations (compared to DBL) changes with respect to numerical choices (such as the order of numerical schemes)? If not, do they expect similar quality?

**Reply:**

In response to this question, we have conducted additional experiments. We adopted the passive transport test with a Hadley-like meridional circulation (Kent et al. 2013) to specifically investigate the impact of reduced precision on various nominal-order advective flux operators.

We tested a range of icosahedral grid resolutions (G5-G9, approximately 240 km to 15 km) and employed different nominal-order ($2^{nd}$, $3^{rd}$, $4^{th}$) horizontal advective flux operators. The results indicate that, regardless of the chosen horizontal flux operator, the single-precision simulations are largely comparable to the double-precision simulations across all resolutions, exhibiting nearly identical convergence rates. The findings from this test support our optimized results, indicating that the advective components of the equations are not significantly sensitive to the precision level, irrespective of whether higher or lower-order operators are utilized.

It would be nice to know if the need for specific terms in double precision comes from the underlying physics and PDEs, rather than from the particular details of the numerical scheme.

The complexity of determining the necessity for double-precision terms spans across multiple levels: from the fundamental physical laws yielding the raw PDEs, to a specific model formulation (like GRIST), and finally to the particular code implementation of that model (like this version). So it is not easy to clearly say which one is determining the outcome.

Safely speaking, our perspective is that the current optimization outcome is primarily relevant to the model and code under consideration and may not be universally applicable across all models. This assertion is easy to verify because there are already several dynamical cores using nearly-pure single precision (e.g., Váňa et al. 2016; Nakano et al. 2018) without performance loss.

On the other hand, the outcome implies that, within the current implementation, the advective part of the model demonstrates greater resilience when subjected to changes in precision. We may also argue that one can more confidently adopt reduced-precision for the advective part, while the pressure-related and some other terms may require more careful code implementation. This is an inference but can be a reference for other models.

The relative vulnerability of the pressure-related term may have some potential sources. First, pressure-gradient terms tend to suffer from the cancellation of significant digits, as also outlined by Nakano et al. (2018). Meanwhile, pressure-related terms are more tightly related to the fast processes in the atmospheric dynamics, and thus they could potentially amplify undesiable small pertubations. Advective process is relatively slower.

Looking ahead, we aspire for a further optimization and code modifications to achieve more reduced runtime while preserving the robustness of the physical performance (e.g., stability, accuracy, convergence).

8. While single precision is definitely more appealing at km-scale, it can still be interesting to use it at coarser resolutions. For instance, it could allow running larger ensembles, benefiting UQ investigations. The authors mention the G6, G7, and G8 grids, which are all km or sub-km grids. Have they done any experiment at lower resolutions? If so, did they observe similar patterns? If they haven't done such experiment, are they planning to? Why or why not?

**Reply:**

This question is referring to the splitting supercell thunderstorms test in Section 4.2. In this particular experiment, which was conducted on a reduced Earth radius, we employed the G6, G7, and G8 grids, corresponding to horizontal resolutions of approximately 1 km, 0.5 km, and 0.25 km, respectively.

To further explore the effects of resolution on precision, we incorporated additional tests at coarser resolutions using the G4 grid (~4 km), and detailed descriptions of these tests have been added to the revised Section 4.2. At this coarser 4km resolution, we observed that the differences between mixed-precision and double-precision simulations were significantly smaller than those noted in the km or sub-km scale

simulations, aligning with earlier experiences.

From 1 km to higher sub-km resolutions, the differences do not monotonically increase with finer resolutions. This phenomenon likely stems from the design of the DCMIP2016 supercell test, which is intended to ensure converged solutions as resolution increases (Klemp et al. 2015). The diminished precision sensitivity at 0.25km is likely related to this convergence.

*Please note that in the original manuscript, the horizontal section shown in Figure 3 was not actually interpolated to a height of 5 km. We have corrected this mistake in the revised manuscript.*

9. In terms of reproducibility, it would help if the authors could share a snapshot of the source code repo, containing all the needed modifications. It would also help to share (perhaps in the form of README files in that same repo) instructions on how to run the particular experiments they ran (e.g., input files, run scripts, peculiar environment settings,...).

**Reply:**

In the revised manuscript, Figure 1 now illustrates how the code has been adapted to facilitate mixed-precision computing. Additionally, we've included Section 2.4, which offers a concise overview of the major modifications made to the original code repo.

For all the test cases addressed here, the corresponding code, build and runtime configurations, as well as necessary input data have been made publicly available on Zenodo. The readers can replicate these tests within their own local environments.

Again, it is emphasized that the computational savings reported may vary based on different configurations, such as grid resolution, computing environment, and compiler settings, among others.

**Reference:**

Kent, J., P. A. Ullrich, and C. Jablonowski, (2013), Dynamical core model intercomparison project: Tracer transport test cases. *Quarterly Journal of the Royal Meteorological Society*, 140(681)**,** 1279-1293.doi:10.1002/qj.2208.

Klemp, J. B., W. C. Skamarock, and S. H. Park, (2015), Idealized global nonhydrostatic atmospheric test cases on a reduced-radius sphere. *Journal of Advances in Modeling Earth Systems*, 7(3)**,** 1155-1177.doi:10.1002/2015MS000435.

Nakano, M., H. Yashiro, C. Kodama, and H. Tomita, (2018), Single Precision in the Dynamical Core of a Nonhydrostatic Global Atmospheric Model: Evaluation Using a Baroclinic Wave Test Case. *Monthly Weather Review*, 146(2)**,** 409-416.doi:10.1175/MWR-D-17-0257.1.

Váňa, F., P. Düben, S. Lang, T. Palmer, M. Leutbecher, D. Salmond, and G. Carver, (2016), Single Precision in Weather Forecasting Models: An Evaluation with the IFS. *Monthly Weather Review*, 145(2)**,** 495-502.doi:10.1175/MWR-D-16-0228.1.

---

## Author Comment (AC2)

**General comments:**

In this paper authors describe the effort associated with making the GRIST dynamical core computationally cheaper through adopting reduced precision to selected parts of the code. The paper is really well written, logically well structured which helps its readability. Like that it is a joy to be followed for a reader. The thorough evaluation part is quite impressive as it represents a lot of hard work. It is also done with great care to cover all possible aspects potentially impacted by reduced precision.

**Reply:**

We would like to express our gratitude to Dr. Filip Váňa for detailed review and constructive feedback on our manuscript. We appreciate your insightful comments and suggestions. Each issue you highlighted has been thoroughly addressed in our revised manuscript. Your expertise has been essential in enhancing both the quality and clarity of our work.

The only complaint is that the paper doesn't really attempt to modify the original code in order to make it more profitable for reduced precision. By that I mean that authors were only trying to identify precision sensitive parts requiring to be exclusively evaluated with double precision in the original code. There is no discussion trying to explain this sensitivity neither an attempt to eventually propose a modification or new method allowing the ussage of single precision also there. But that is perhaps subject to another paper.

**Reply:**

Indeed, the focus of this work has been on identifying precision-sensitive parts of the original code, without delving into further modifications of complex or "tricky" code segments. This work could serve as an initial step towards the broader application of reduced precision within the GRIST dynamical core. Future developments could include rewriting suboptimal code implementations and exploring the use of half-precision computations, which could further enhance computational efficiency and resource utilization.

**Small points:**

1. In 110 some error norm is computed based on two model variables Ps and VOR. How those norms are evaluated and is there any scaling applied to one of them to make the two norms roughly comparable? The way it is described here is too generic to be followed.

**Reply:**

In this manuscript, Eq. (2) evaluates the error norms for two model variables: surface pressure (Ps) and relative vorticity (VOR). The error norms are defined as follows:

$$L_1 = \frac{I(|\mathcal{H} - \mathcal{H}_T|)}{I(|\mathcal{H}_T|)} \tag{1}$$

$$L_2 = \sqrt{\frac{I[(\mathcal{H} - \mathcal{H}_T)^2]}{I[(\mathcal{H}_T)^2]}} \tag{2}$$

$$L_\infty = \frac{max\forall|\mathcal{H} - \mathcal{H}_T|}{max\forall|\mathcal{H}_T|} \tag{3}$$

$$I(\mathcal{H}) = \int_{z=z_{surface}}^{z=z_{top}} \oiint \mathcal{H}\ dA dz \tag{4}$$

where $\mathcal{H}$ and $\mathcal{H}_T$ are the computational solution and true solution, respectively. Max $\forall$ means select the maximum value from the field. $I(\mathcal{H})$ denotes global 3D integration for an arbitrary quantity $\mathcal{H}$, $A$ denotes cell area and $z$ denotes height. Vertical integral will be omitted for a 2D integration.

Your observation is accurate. As you noted, the error magnitude associated with pressure (Ps) is smaller compared to vorticity (Vor). Consequently, according to our current criteria (defined beforehand), only "Vor" significantly influences our optimization outcome. This point has been clearly stated in the revised manuscript.

For the baroclinic wave test, "Vor" demonstrates a higher sensitivity to small perturbations than other physical variables. This heightened sensitivity makes it a good metric in our optimization procedure.

2. I found bit unintuitive to digest the results of splitting supercell thunderstorms in section 4.2. Especially, the text belonging to 245 part describing results presented on figure 4. I am bit surprised by the great similarities between double precision and mixed solution until the 5400s with almost bifurcation behaviour afterwards. It feels like something strange happens at that time range.

**Reply:**

The revised manuscript's Figure 5 provides a detailed depiction of the temporal evolution of the domain maximum vertical speed and area-integrated rainfall rate in the supercell thunderstorms simulation. During the initial 5400 seconds of the simulation, the results from the mixed-precision and double-precision simulations are quite comparable. This similarity can be attributed to the relatively weak vertical motions present before the supercell reaches maturity, which are not highly sensitive to the level of precision used in the computations.

After 5400 seconds, as the supercell matures and develops more complexity, including the generation of small-scale features (now Fig. 4). As the simulation progresses, these small-scale perturbations amplify. Therefore, as the storm develops and becomes more dynamic, the sensitivity to the precision level becomes more pronounced, but still limited to relatively small scales.

I am also quite surprised by finding the highest resolution runs continue to remain similar across the two precisions while lower resolution runs show difference. From our experience it was rather opposite: higher resolution runs exhibited higher sensitivity to used numerical precision. Could this be somehow explained?

**Reply:**

In our revised manuscript, we have added tests at G4 (~4 km) grid. We then show the differences between mixed-precision and double-precision simulations at heights of 2.5 km, 5 km, and 10 km, respectively. It is found that at 4 km, the differences between mixed-precision and double-precision simulations are much smaller than at other resolutions, for all examined heights (Figs. 4b-c). As the resolution increases from 4 km to 1 km, the differences in the supercell become more pronounced (Figs. 4b-d, f-h, j-l). The contrasting behaviours between 4km and 1km are consistent with earlier

experience and research studies.

As you noted, the differences between mixed-precision and double-precision simulations do not increase monotonically as resolution moves from 1 km to 0.5 km and to 0.25 km. Instead, the differences diminish. The design of the DCMIP2016 supercell test is intended to ensure converged solutions as resolution increases. At 0.25 km, the mixed-precision simulations are closer to the double-precision results at all heights (Figs. 4n-p). This is likely related to the solution convergence.

*Please note that in the original manuscript, the horizontal section shown in Figure 3 was not actually interpolated to a height of 5 km. We have corrected this mistake in the revised manuscript.*

Despite my general comment and the two rather questions than really complain I would suggest the paper is accepted for publication. If author wish they could eventually address my points, but it could be published straight away the way as it was submitted. Bravo!

Filip Vana (ECWMF)

**Reply:**
Thank you! Dr. Váňa.

---

## Author Response (AR2)

**Topic editor decision: Publish subject to minor revisions (review by editor)**
by Peter Caldwell
**Public justification (visible to the public if the article is accepted and published)**:

**Like the reviewers, I thought this was a great paper. I also thought the authors did a decent job of addressing the reviewer comments. In my own read-through of the most recent draft, I found a few things that could make the paper even better. Thus I expect the paper will be accepted after some minor corrections.**

**Reply:** We would like to express our gratitude to Dr. Peter Caldwell for his handling, detailed review, and constructive feedback on our manuscript. We appreciate your insightful comments and suggestions. The issues you highlighted have been addressed in the revised manuscript. They help to further improve the clarity and quality of the paper.

1. **I was a bit disappointed that many of the authors' reviewer responses were just that - they didn't actually change the text to address the question. This is unfortunate because many readers will have the same questions. I was also a bit disappointed that the authors just conjectured in a lot of these cases rather than actually doing any analysis. I think the paper is in great shape as-is so I don't insist that more analysis is done, but I thought some of the reviewers' questions were quite good. I'm thinking particularly of reviewer 1 questions 1-2, 4, and reviewer 2 question 1.**

**Reply:** We acknowledge that these questions warrant more detailed investigation. The reviewers' questions have been now discussed in the main text. Specifically, we **have revised lines** 119-122, 407-413, and 200-224 to **address** Reviewer 1's questions 1-2 and 4, and lines 122-125 to **address** Reviewer 2's question 1.

2. **In lines 16-17 of the abstract, the authors refer to the \*solver\* for hydrostatic, non-hydrostatic, and advection but I think they mean that these are timing reductions for the whole dycore (or the advection part). In particular, they say later that the dycore solver is actually slower in mixed precision. Another issue is that around line 186 the timings are described for the \*dry\* dycore but the moist dycore is used for most of the physics tests. Is there a difference in timing?**

**Reply:** Yes, in lines 16-17 of the abstract, the hydrostatic, non-hydrostatic, and advection solvers refer to the timing reductions for the whole dycore (in our term, dycore means dry dynamical core, not with tracer transport) or the advection part (tracer transport). The mixed precision dycore solver is slower only when compared to the single-precision dycore solver, because in the mixed precision dycore solver, the precision-sensitive calculations remain in double precision. This is described around lines 195-196 of the manuscript.

The computational time is diagnosed independently for the dycore (hydrostatic or nonhydrostatic) and tracer transport module, i.e., 24%, 27%, 44% in the abstract. The physical performance (Section 4.1-4.5) focuses on the moist dynamical core (dycore + tracer). This is to give a clean comparison.

**I'd also like to know how much faster the full model is when the dycore uses mixed precision for the dycore, both in AMIP and global storm-resolving**

**model (GSRM) configuration. Since these simulations were used for physics analysis, it should be trivial to report their timings. In my own GSRM, we spend ~90% of our run time in the dycore so I suspect GRIST total performance improvement will be O(20%) in this case.**

**Reply:**

We have now mentioned "how much faster the full model is" in the AMIP and GSRM simulations. The GSRM tests have an impressive time reduction. Your estimation (~20%) is rather accurate. The actual number derived from the DBL and MIX tests in Section 4.5 was 25.5%, diagnosed for the dynamics and physics procedures, including physics-dynamics coupling.

The AMIP test was performed at 120-km resolution using 640 MPI processes distributed across 32 nodes. The MIX code did not witness very impressive time reduction (~12% for the hydrostatic model), because the memory-bound issue may not be significant for such low-resolution simulations.

**This is further detailed at the 3$^{rd}$ and 4$^{th}$ paragraphs of Section 3.**

3. It would be nice if the authors could say whether this project has been deemed a success internally: will all future users be told to use the mixed-precision mode, or do you think the single- versus double precision differences are too big to be used for operations and/or research?

   As an aside, I was surprised this paper just looks at differences between simulations rather than checking whether model skill is significantly affected (as in https://rmets.onlinelibrary.wiley.com/doi/full/10.1002/qj.4181, for example). If mixed-precision isn't good enough, would being more stringent in deciding which variables and equations needed double precision have resulted in a more acceptable solution (and is this easy enough that the authors could try this if need be)?

**Reply:** We consider this project to be a success, at least in its current phase. From idealized experiments and AMIP simulations to GSRM simulations, the mixed-precision simulations have well reproduced solutions as the double-precision simulations and saved the computational time. Based on these test cases, the mixed-precision model code can be said largely successful and has been used for many ongoing efforts.

Safely speaking, for quality operational runs, more testing efforts are required. The examination may further include checking operational metric, skill score card, etc, as the paper you mentioned. This requires longer term efforts.

Mixed-precision computing will maximize its value in the kilometer-scale applications, and future applications and studies will more focus on this scale, including metric and skill score related to the k-scale forecast. One plan is to merge the mixed-precision code and a limited-area model recently developed for GRIST, to enable efficient and skillful customized model applications at the convection-permitting scale.

**We have added a paragraph at the end of Section 5 to address this issue.**

4. Around line 110, it might be useful to clarify that error is calculated relative to the double-precision results (rather than observations).

**Reply:** We have **clarified** in the manuscript that the error calculations are relative to the double-precision results.

5. typos (please do one more careful read through before you resubmit!):

a. L253: "appears, expectED for..."

b. L320: "may DIVERGE MORE from..."

c. L354: "a few grid spacES"

d. L397: "It also needs to BE recognizeD that"

e. Fig 7 caption: fix "from the and deterministic"

f. Fig 9 caption: I think you mean the continuity equation rather than the continuous equation.

**Reply:** The typos mentioned have been individually **corrected**. We have **carefully read** the manuscript again to avoid potential mistakes. We have also slightly improved some sentences to improve the clarity of the paper. Thank you!